# Could the type and severity of gross lesions in pig lymph nodes play a role in the detection of *Mycobacterium avium*?

Aleksandra Kaczmarkowska[1]☯*, Anna Didkowska[1‡], Sylwia Brzezińska[2‡], Daniel Klich[3‡], Ewelina Kwiecień[4‡], Izabella Dolka[5‡], Piotr Kociuba[6‡], Magdalena Rzewuska[4‡], Ewa Augustynowicz-Kopeć[2‡], Krzysztof Anusz[1‡]

1 Department of Food Hygiene and Public Health Protection, Institute of Veterinary Medicine, Warsaw University of Life Sciences (SGGW), Warsaw, Poland, 2 Department of Microbiology, National Tuberculosis Reference Laboratory, National Tuberculosis and Lung Diseases Research Institute, Warsaw, Poland, 3 Department of Animal Genetics and Conservation, Institute of Animal Sciences, Warsaw University of Life Sciences (SGGW), Warsaw, Poland, 4 Department of Preclinical Sciences, Institute of Veterinary Medicine, Warsaw University of Life Sciences (SGGW), Warsaw, Poland, 5 Department of Pathology and Veterinary Diagnostics, Institute of Veterinary Medicine, Warsaw University of Life Sciences (SGGW), Warsaw, Poland, 6 Department of Environmental Protection and Landscape Preservation, Institute of Mathematics, Informatics and Landscape Architecture, The John Paul II Catholic University of Lublin (KUL), Lublin, Poland

☯ These authors contributed equally to this work.
‡ AD, SB, DK, EK, ID, PK, MR, EAK and KA also contributed equally to this work.
* aleksandra_kaczmarkowska@sggw.edu.pl

**Data Availability Statement:** All relevant data are in the manuscript and also available upon request to the Ethics committee (lke@sggw.edu.pl).

## Abstract

The *Mycobacterium avium* complex (MAC) comprises a widespread group of slowly-growing bacteria from the *Mycobacteriaceae*. These bacteria are responsible for opportunistic infections in humans and animals, including farm animals. The aim of the study was to determine whether it is possible to predict the presence of *M. avium* in pig lymph nodes based on the size and type of lesions found during post-mortem examination at a slaughterhouse. Lymph nodes were collected from 10,600 pigs subjected to such post-mortem examination. The nodes were classified with regard to their quality, and the number of tuberculosis-like lesions; following this, 86 mandibular lymph nodes with lesions and 113 without visible macroscopic lesions were selected for further study. Cultures were established on *Löwenstein-Jensen and Stonebrink media, and a commercial* GenoType Mycobacterium CM test was used to identify and differentiate *M. avium* species. The prevalence of *M. avium* was 56.98% in the lymph nodes with lesions and 19.47% in the unchanged ones. Statistical analysis indicated that visual assessment of lesions in the mandibular lymph nodes, in particular the number of tuberculous lesions, is a highly-efficient diagnostic tool. Similar results were obtained for estimated percentage area affected by the lesion, i.e. the ratio of the changed area of the lymph node in cross-section to the total cross-sectional area of the lymph node; however, this method is more laborious and its usefulness in slaughterhouse conditions is limited. By incising the lymph nodes and assessing the number of tuberculosis-like lesions, it is possible to limit the inclusion of meat from pigs infected with *M. avium* into the human food chain.

**Funding:** The authors received no specific funding for this work.

**Competing interests:** The authors have declared that no competing interests exist.

## Introduction

The *Mycobacterium avium* complex (MAC) is a group of globally-occurring opportunistic bacteria with zoonotic potential that are transmitted mainly by livestock and birds. MAC infections in humans represent an increasingly pressing problem, and there is a need for special measures intended to prevent their spread [1]. The complex is responsible for various diseases, including localized infections such as cervical lymphadenitis in immunocompetent patients, and disseminated pulmonary infections in immunodeficient ones [2–4].

Some of the most commonly-occurring species in pigs are *M. avium* ssp. a*vium*, *M. avium* ssp. *hominissuis*, *M. avium* ssp. *silvaticum*, and *M. avium* spp. *paratuberculosis* [5]. They infect mainly the respiratory and intestinal tract, and lead to economic losses among pig farmers [6, 7].

According to the United Nations Food and Agriculture Organization, pork remains the most widely-consumed meat in the world. As such, it is critically important to ensure it is of the highest microbiological quality. This poses a significant challenge, as the pathogenesis of MAC infection in pigs remains not fully understood [8, 9]. Carcasses are typically subjected to post-mortem examination, during which, attention should be paid to the presence of gross lesions suggesting mycobacteria infection. These lesions can be detected in lymph nodes, most commonly intestinal ones, in which they take the form of tuberculosis-like lesions [8]. In both humans and pigs, the predominant target organ of MAC is the liver [10, 11].

During the morphological inspection of carcasses at slaughterhouses, tuberculosis-like gross lesions are most commonly observed in the lymphatic organs, such as the mesenteric lymph nodes, and less often in the mandibular lymph nodes and Peyer´s patches or tonsils [1]. It is suspected that those lesions frequently go unrecognized [12], and some molecular detection solutions are being introduced for foodborne risk assessment [13].

Current European Union legislation recommends visual inspection of the lymph nodes without incision [14], which makes it impossible to identify potential pathological lesions. This three-stage post-mortem examination increases the chance of identifying lesions in the lymph nodes, which is one of the objectives of the examination. Purely visual examination, without incision, runs the risk of missing pathological lesions, which can enter the human food chain. Minced meat, with the addition of lymph nodes, seems to be particularly risky as a potential source of human infection with *M. avium* [15].

Thanks to their resistance to unfavorable conditions, mycobacteria can survive some disinfection or decontamination procedures used in food production, as well as thermal processing. Many studies have reported detection of mycobacteria in animal tissues and milk, thus confirming that food of animal origin can be a source of mycobacterial transmission to humans [16]. Research has also confirmed the presence of mycobacterial DNA and live bacteria in commercially-available food products such as raw pork, fermented salami-type products and heat-treated roast meat [17]. Considering the potential threat to public health posed by infected food, the aim of this study was to determine whether it is possible to predict the occurrence of *Mycobacterium avium* in swine lymph nodes based on the type and size of gross lesions found on them. An additional aim was to determine the frequency of tuberculosis-like lesions and atypical mycobacteria in pigs in the study area and assess whether there is a relationship between the incidence of *M. avium* and the sex of animals and the size of the farms where the animals were kept.

## Materials and methods

### Material

A total of 10,600 pig carcasses were subjected to post-mortem examination including the incision of submandibular and mesenteric lymph nodes. In 86 animals, anatomical lesions were

found in the mandibular lymph nodes but not in the mesenteric lymph nodes. The mandibular lymph nodes with lesions (n = 86) and a randomly-selected group of mandibular lymph nodes without lesions (n = 113) were then tested for the presence of *Mycobacterium* spp. The animals from which the material was collected were aged between 5.5 months and four years (mean age: 6.4 months). Of the 199 carcasses, 99 were male and 100 were female. The animals came from 83 farms located in central Poland, ranging in size from 2 to 3480 individuals; mean size 117 animals. The harvested mandibular lymph nodes were kept in a freezer at -20˚ C before analysis.

No ethical approval was required from the National Ethical Committee for Animal Experiments to conduct the tests because the lymph nodes were collected as a part of the post-mortem inspection of pigs in the slaughterhouse. No animal was intentionally sacrificed for this study.

## Lesion classification and histopathology

The collected lymph nodes were all classified into three groups based on the type of lesion, and three groups based on the number of lesions; the classification was performed by a veterinarian with eight years of experience in a slaughterhouse. The nodes in which no macroscopic lesions were found were assigned to Group 0, those with pinhead-size, nodule-like individual lesions were placed in Group 1 (Fig 1), and those with numerous pinhead-size or diffuse lesions into Group 2 (Figs 2 and 3).

The collected material was also classified according to the nature of the lesions: group A—non-purulent lesions (Figs 1 and 2), group B—purulent lesions (Fig 3), group C—no lesions. A

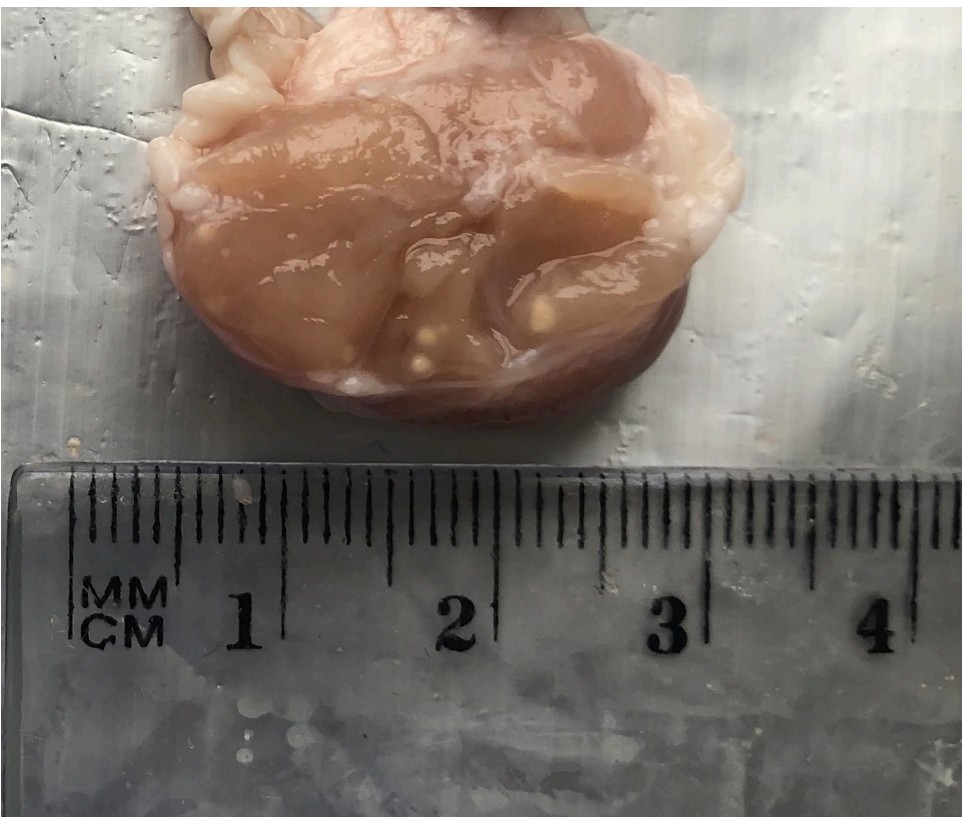

**Fig 1. Mandibular lymph node with single lesions, non-purulent.**

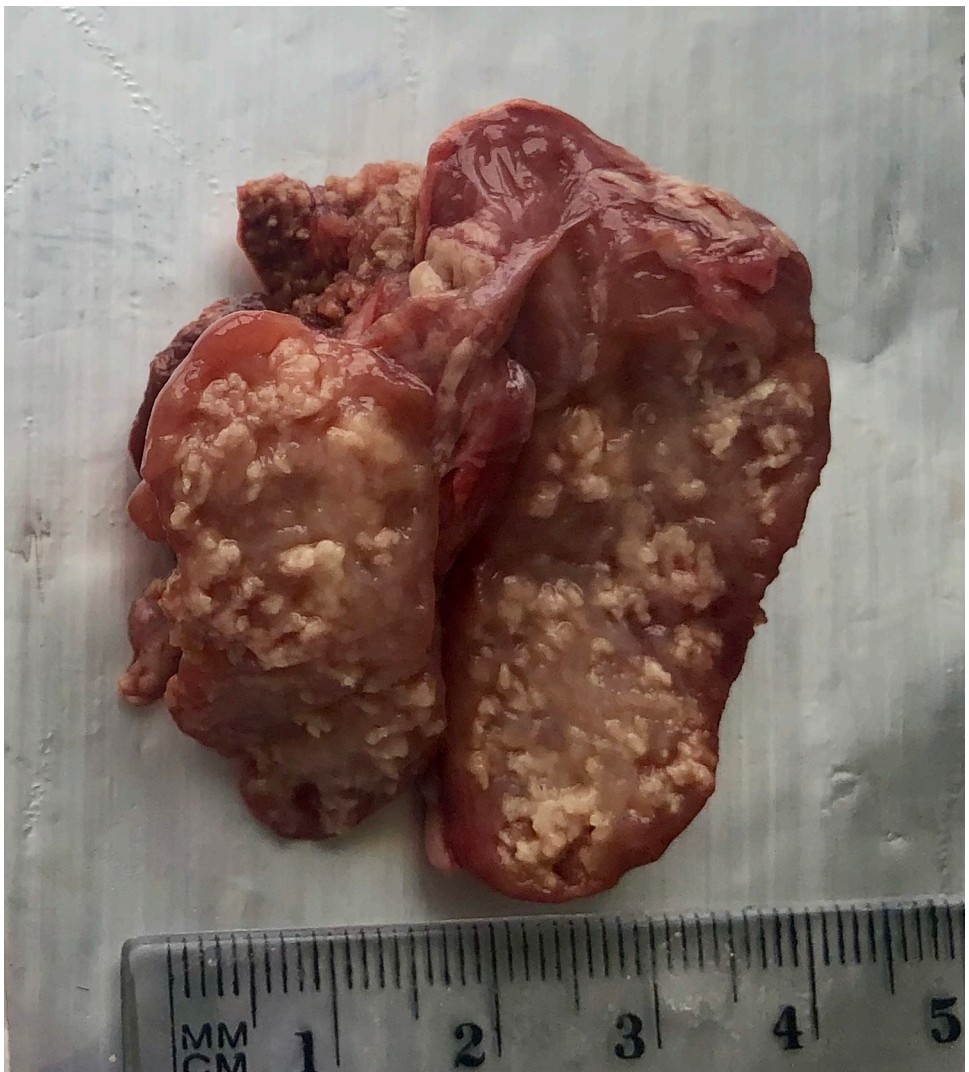

**Fig 2. Mandibular lymph node with numerous pinhead-size lesions, non-purulent.**

photogrammetric examination of the lymph nodes was also performed; briefly, the changed area of the node in cross-section was measured by image analysis and this was then expressed as a percentage of the entire cross-sectional area of the lymph node. Fragments of mandibular lymph nodes approximately 30 x 30 x 15 mm in size were collected into 4% formalin for histopathological examination. The histopathological examination only included a few of the selected lymph nodes with apparent lesions, i.e. some from groups A, B, 1 and 2: the nodes from groups C and 0 were not examined. The histopathological testing itself was based on standard hematoxylin and eosin staining to assess lymphocytic infiltration and Kinyoun staining to detect mycobacteria.

## Mycobacterial isolation

Acid-fast mycobacteria were isolated from the material according to the recommendations of the Reference Microbiological Laboratory of the National Research Institute—National Veterinary Institute in Puławy, Poland. The material was pre-minced with sterile scissors and placed

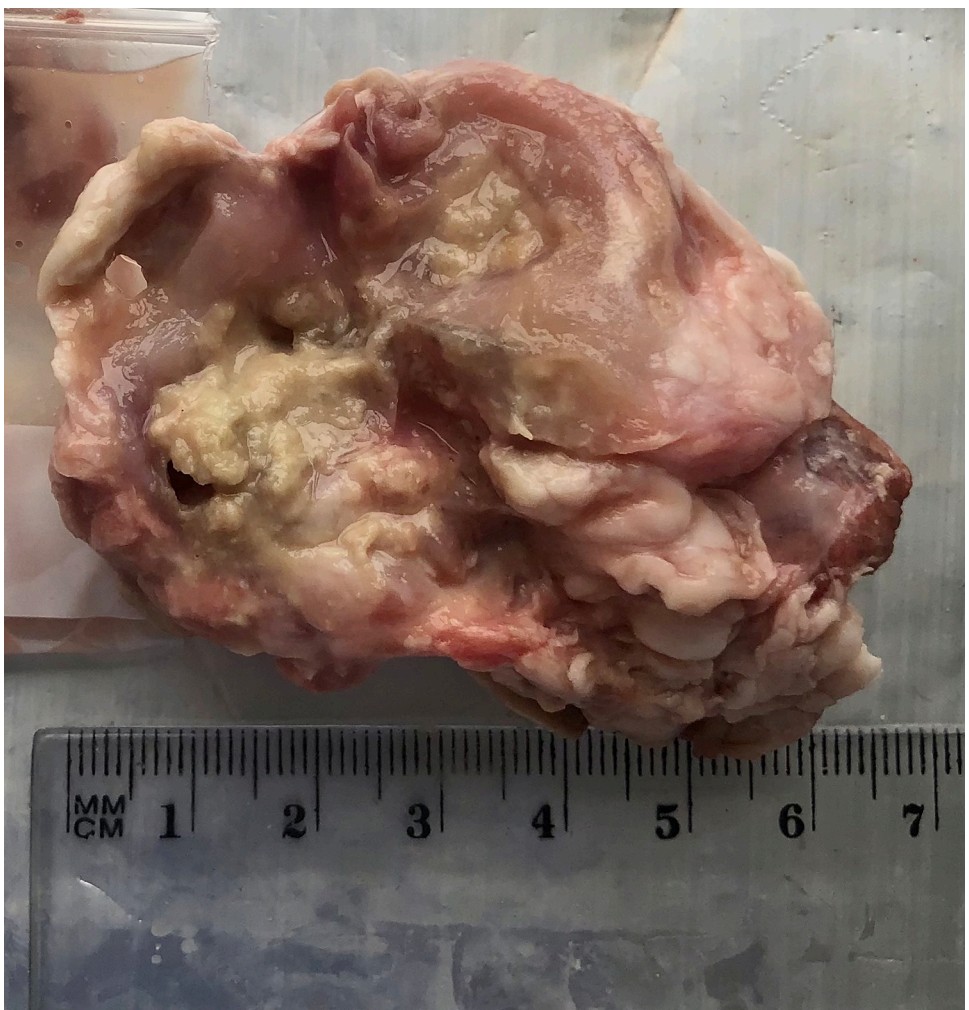

**Fig 3. Mandibular lymph node with numerous pinhead-size lesions, purulent.**

in bags with a filtering membrane (BagPage® 100). The material was then submerged in a 5% solution of oxalic acid (POCH, Poland) for decontamination and homogenized with a stomacher for three minutes at 12 strokes/second. The resulting suspension was poured into tubes. These were incubated for 10–15 minutes at 37˚C, and then centrifuged for 10 minutes at 1500 x g. In the next step, the supernatant was removed, and sterile 0.9% NaCl was added to the maximum volume of the tube. The tubes were shaken and centrifuged for 10 minutes at 1500 x g. This operation was repeated twice. The sediment (5μl) was used for inoculation on solid media for mycobacterial cultures: Löwenstein-Jensen (MERCK, Germany) and Stonebrink (MERCK, Germany). The cultures were incubated at 37˚C for 12 weeks. The media were checked for mycobacterial colony growth once a week. No growth after 12 weeks was considered a negative result.

## DNA isolation

The loop of an inoculation loop filled with mycobacterial colonies isolated on Löwenstein-Jensen or Stonebrink media was suspended in 150 μl of water, and then incubated for 30 minutes in a heating block at 95˚C. After incubation, the tubes were centrifuged for five minutes at 15000 x g. The supernatant was transferred into new tubes and used for testing.

## Genotyping

*Mycobacterium avium* species were identified and differentiated using a commercial Geno-Type Mycobacterium CM test (Hain Lifescience, Germany). The isolated DNA was selectively replicated by polymerase chain reaction (PCR). The resulting amplicons were transferred onto DNA strips covered with highly-specific probes which were complementary to the amplified DNA sequences. The amplicons were bound to their complementary sites, while the unbound fragments were removed during washing. Following this, a conjugate labeled with alkaline phosphatase was added, and the mycobacterial species were identified based on the hybridization pattern of specific probes placed on the strips with the product of the multiplex PCR reaction. The result was read using a dedicated template attached by the manufacturer.

## Statistical analysis

To identify the factors that can explain the presence of *M. avium*, a generalized linear binary model was employed. In the model, the presence of *M. avium* in the samples was used as a dependent variable. Each sample where the presence of *M. avium* was confirmed was marked as 1, while those where no *M. avium* presence was confirmed were marked as 0. The following independent variables were included in the analysis: 1) qualitative severity of the lesions in the lymph nodes, 2) quantitative severity of the lesions in the lymph nodes, 3) sex of the pigs, 4) herd size. The qualitative severity of the lesions fell into three categories: A–non-purulent lesions, B–purulent lesions, C–no lesions. The quantitative severity of the lesions fell into three categories: 0 –no lesions, 1 –single lesions, and 2 –numerous lesions. To find the best fit model, all model variants were run (i.e. the variables were included in all possible combinations, including the null model), and the models were compared using Akaike Information Criteria (AIC) according to Burnham and Anderson [18]. The model with the lowest AIC value was regarded the best fit. When the AIC difference was below 2, the simpler model (i.e. with lower k) was selected according to the *Ockham's razor rule*. A *post hoc* pairwise comparison of the frequency of *M. avium* in lymph nodes of given category was performed with the least significant difference test (LSD).

A logistic regression model was also run to verify whether the presence of *M. avium* in the samples could be predicted from the percentage of lesions in the lymph nodes. In the model, the presence of *M. avium* in the samples was the dependent variable and the percentage of lesions was the independent variable. The dependent variable was similar to that used in the generalized linear binary model.

The percentage of the lesion area was calculated using AutoCAD 2020 software. Photos of the lymph nodes were taken with a Sony Cyber-shot DSC-W830 camera with a linear scale, which was then used to scale the images to 1:1 scale. The surfaces of the lymph nodes and lesions were marked with a spline tool. For each lymph node, the total cross-sectional area and the area occupied by the lesions (in $cm^2$) were calculated with the Area tool. To verify whether percentage cover of lesions or quantitative severity of lesions in lymph nodes can better predict the presence of *M. avium* in the samples, a similar logistic regression model was run on the same set of observations; the quantitative severity of lesions was used as the independent variable in both models. Both models were compared according to $R^2$, classification tables and AUC (area under the ROC curve).

## Results

### Lesion classification and histopathology

The lymph nodes collected for the study were classified into the following groups 0 (n = 113), 1 (n = 38), 2 (n = 48), A (n = 73), B (n = 13), and C (n = 113). Affected lymph nodes presented

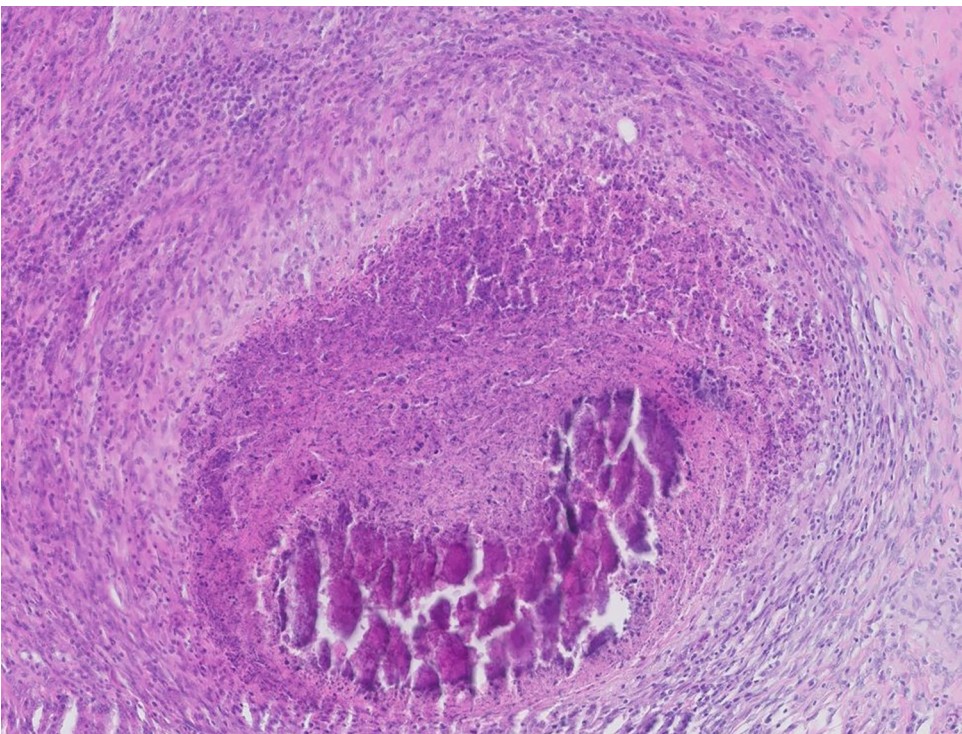

**Fig 4. Histopathologic examination of mandibular lymph node.** The nodes presented multifocal to coalescing coagulation necrosis, with areas of mineralization. The necrosis was separated by a thick layer of proliferating fibrous tissue. Sinusoids contain a large number of macrophages, partly with a granular brownish cytoplasmic pigment. Multifocally prominent lymph follicles are evident, with a large germinal center and prominent parafollicular lymphocyte proliferation.

chronic, multifocal to diffuse, granulomatous or purulent-granulomatous inflammation with polynuclear giant cells *(Fig 4)*. The majority of the lymph nodes contained thrombotic necrosis with mineralization or lytic necrosis, surrounded by fibrous proliferation. Sometimes *bodies similar to the Splendore-Hoeppli phenomenon were also detected, represented by luminous eosinophilic material (Fig 5)*.

The incidence of tuberculosis-like lesions in pig lymph nodes was found to be 0.81%: i.e. 86 lymph nodes with tuberculosis-like lesions out of 10 600 examined carcasses.

## Mycobacterial isolation

Of the samples taken from affected lymph nodes, bacterial growth was observed in 63 samples cultured on Löwenstein-Jensen medium, and in 36 samples on Stonebrink medium. Bacterial growth on both media was detected in 31 cases. For the lymph nodes without macroscopic lesions, bacterial growth on Löwenstein-Jensen or Stonebrink media was observed in 58 cases: Löwenstein-Jensen in 29 cases, Stonebrink medium in 29 cases, and for both media in 15 cases.

## Genotyping

The incidence of *M. avium* was found to be 56.98% in the lymph nodes with lesions, and 19.47% in those without. The results are presented in Table 1.

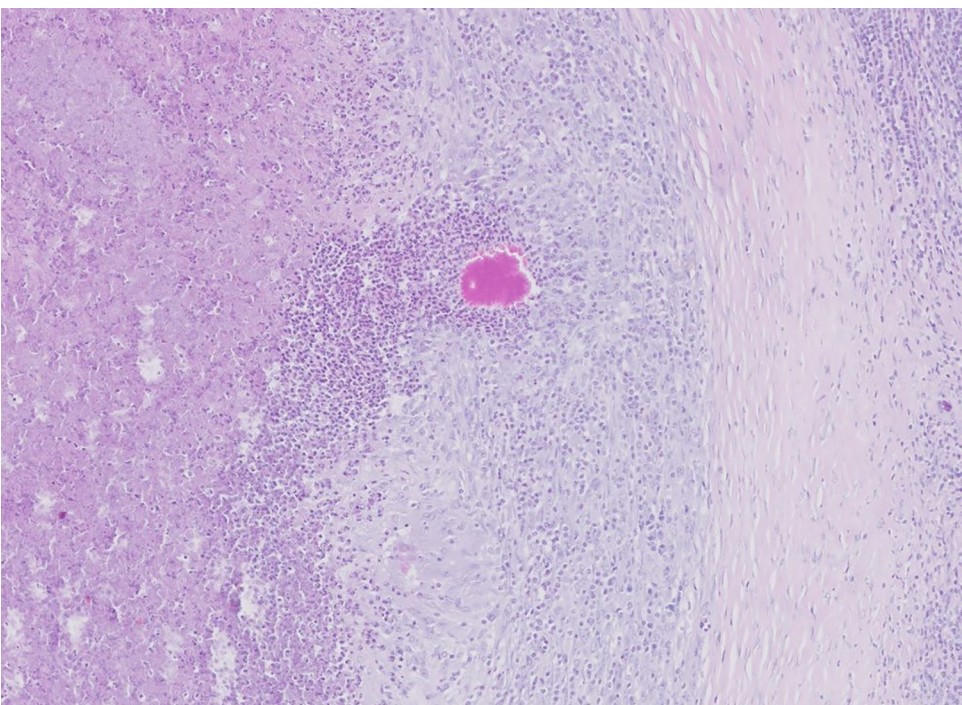

**Fig 5. Histopathologic examination of mandibular lymph node with Splendore-Hoeppli-like material.** Multifocal to coalescing granuloma formation can be seen. Granulomas consist of a central area of necrosis surrounded by a thick rim of partially epithelial macrophages and sometimes multinucleated giant cells. A visible demarcation formed by collagen-rich connective tissue is also present. Splendore-Hoeppli material can be observed, represented by a radiating club-shaped eosinophilic material.

## Statistical analysis

The model selection resulted in the removal of all variables except for quantitative severity of lesions in lymph nodes. Neither the qualitative severity of lesions in lymph nodes, sex of the animals nor herd size significantly explained the presence of *M. avium*. The best model, based on quantitative severity of lesions in lymph nodes, was, however, statistically significant ($\chi^2$ = 46.88, p<0.001). The severity of the lesions was significantly associated with the presence of *M. avium*. In addition, the categories "numerous lesions"and "single lesions" were associated with a significantly higher chance of *M. avium* than the"no lesions" category (Table 2): *M. avium* was 2.5 times more likely to be present in the lymph nodes with single lesions compared to nodes without lesions, and over 13 times higher in the lymph nodes with numerous lesions.

In the "no lesions" samples, the frequency of *M. avium* was 0.23, indicating that *M. avium* could still be present even if there were no lesions, but the probability was low. The frequency of *M. avium* in the "no lesions" samples differed significantly from the other two categories (p = 0.020 in comparison with "single sessions" and p<0.001 in comparison with "numerous

**Table 1. Number of submandibular lymph nodes from which *M. avium* was isolated.**

| Etiological factor | Lymph nodes with lesions | Lymph nodes without lesions |
|---|---|---|
| *M. avium* | 49 | 22 |
| *M. celatum* | 3 | 4 |
| *M. fortuitum* | 1 | 0 |
| *M. avium + M. chelonae* | 3 | 0 |

**Table 2. Effect of qualitative severity of lesions in lymph nodes on the presence of *M. avium* in the generalized linear binary model (* reference category).**

| Source | B | Standard error | Lower CI | Upper CI | p | Exp (B) |
|---|---|---|---|---|---|---|
| Intercept | -1.208 | 0.224 | -1.646 | -0.770 | <0.001 | |
| Quantitative (Numerous lesions) | 2.569 | 0.419 | 1.747 | 3.390 | <0.001 | 13.050 |
| Quantitative (Single lesions) | 0.950 | 0.393 | 0.180 | 1.720 | 0.016 | 2.586 |
| Quantitative (No lesions) | 0* | | | | | |

sessions"). In the "single lesions" category, the frequency was found to be 0.44, which did not unequivocally confirm the presence of *M. avium*. However, this value was 0.8 in the "numerous lesions" category, indicating a high probability of species detection (Fig 6). The "single lesions" and "numerous lesions" categories were also found to differ significantly in the pairwise comparison (p<0.001).

The logistic regression model indicated that the percentage of the lesion area in the lymph nodes could significantly predict the presence of *M. avium* ($\chi^2$ = 15.33, p<0.001). *M. avium* infection was more likely where the lesion covered a greater area of the lymph node (Fig 7): the probability of *M. avium* being present exceeded 0.5 when over 10% of the lymph node area was affected.

The descriptive quantitative severity of lesions and measuring the percentage cover of lesions in lymph nodes yielded fairly similar results. Nevertheless, the model based on the quantitative severity of lesions demonstrated better model fitness: a higher Nagelkerke's r-square value (0.325 and 0.126 respectively), higher percentage of total classified cases (77.1 and 69.4% respectively), and a greater percentage of correctly classified positive observations (56.7

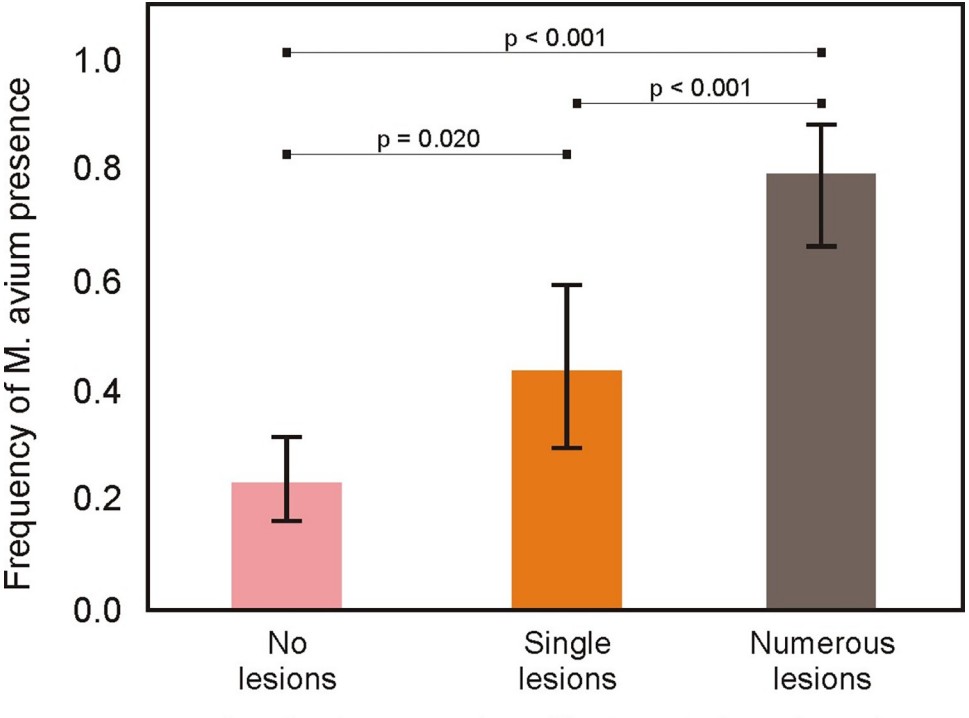

**Fig 6. Frequency of *M. avium* presence (and 95% confidence intervals) with regard to the qualitative severity of lesions in lymph nodes, and pairwise comparison with least significant difference test.**

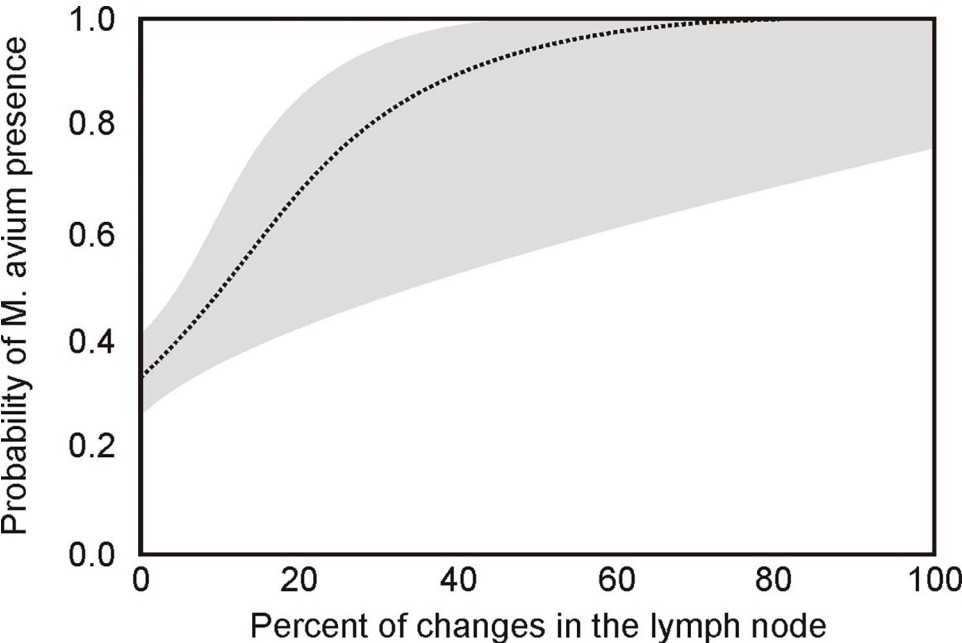

**Fig 7. The probability of *M. avium* presence with 95% confidence intervals (shaded area) depending on the percentage cover of lesions in the lymph node.**

and 23.3% respectively). The AUC, i.e. area under the ROC curve, was similar (0.74 and 0.734 respectively).

## Discussion

Our study confirmed a statistically significant correlation between the size of gross lesions in pig submandibular lymph nodes and the presence of *M. avium*. Taking into consideration the increasing number of MAC infections in humans, and the possibility that the bacteria could enter the food chain, we speculated that a scientific algorithm based on visual inspection that could be carried out in slaughterhouses might be of value in managing the problem. This was confirmed by our statistical analysis, which demonstrated a high probability of isolating *Mycobacterium* spp. from the changed lymph nodes and a low probability of *M. avium* infection in their absence.

In pigs, *M. avium* is known to cause oral infections with localized gross lesions mainly in the head, mediastinum and digestive tract lymph nodes [19, 20]. Based on these findings, and our earlier observations on the frequency of lesions in pigs, we decided to investigate the submandibular lymph nodes, which are easily accessible and take little time to evaluate, i.e. about five seconds.

Our results indicate that preliminary assessment of macroscopic lesions based on visual inspection may be of value in detecting *M. avium*, and that assessing lesion size can be a rapid and cost effective means to achieve this. The only requirement is that the meat examiners receive adequate training. However, this being a new proposal, possibly the first, it does have some limitations. Firstly, regarding the subjective assessment of lesions in lymph nodes, our findings indicate a relatively high probability of *M. avium* being present in lymph nodes with no lesions observed. This indicates the need for a careful approach to visual inspection, and an individual not displaying lesions should not be automatically qualified as free from *M. avium*, but as "probably free". Another limitation is that infection may occur naturally in pigs; as a

result, the lesions observed in the individuals were at different stages of development, or they may not have had time to develop at all. However, this may be an advantage of our methodology, as this method may be used during post-mortem inspection in slaughterhouses that do not receive experimentally-infected animals.

Visual inspection is an efficient screening method, while the sensitivity of other methods, such as cultures or molecular tests, still needs improvement. It also avoids the other limitations of laboratory testing, such as the long time needed to obtain results and the significant financial costs. It seems reasonable to use visual screening to select material for molecular testing, possibly using GenoType Mycobacterim CM. Such improved screening would support public health protection and avoid the great economic losses caused by mycobacterial infections [1]; in addition, detecting a large number of lesions in a specific location can be an important indicator for pig farmers.

Our findings indicate that calculating the lesion area is an efficient prognostic factor for the presence of *M. avium*. However, large areas of tuberculosis-like lesions can also be attributed to co-infection by *M. avium* and *Streptococcus spp.*, *Staphylococcus aureus* or *Rhodococcus equi*. In some cases, it is impossible to isolate and culture any etiological factor from advanced lesions due to their strong calcification and devitalization of mycobacteria [21].

The method used in the present study to calculate the percentage area of a lesion is time-consuming and troublesome and requires specific hardware and software. In addition, it does not provide better results than the visual assessment of the number of the lymph node lesions. Nevertheless, it should be noted that the lower precision of the model regarding the assessment of percentage lesions could have been due to the smaller sample size, as not all the nodes were inspected for the lesions on their surface.

Moreover, in the descriptive quantitative method, the frequency of the "single lesions" category did not reflect the presence of *M. avium*. Only situations where no lesions or numerous lesions were observed proved to be clear indications of the possible presence of *M. avium*. For this reason, it seems justified to use a combination of both methods simultaneously, i.e. lesion number and node coverage, at least for the "single lesions" category. The lesion area can be visually assessed on site. The results of the logistic regression indicate a 50% risk of *M. avium* even when only 10% of the lymph node cross section is affected: single lesions that account for over 10% of the cross section area also demonstrate an increased risk of *M. avium*, which increases further with percentage cover. When 50% of the node section is affected, the presence of *M. avium* is nearly guaranteed. Therefore, estimating the area of the lesions seems a reliable and convenient method for assessing the risk of *M. avium* incidence.

Our histopathological analysis confirmed the presence of necrosis and calcification foci, these being lesions typical of late stage granulomatous inflammation caused by *M. avium* [10]. However, the examination did not reveal any significant differences between materials collected from different types of lesions; this confirms our hypothesis regarding the value of the macroscopic assessment of lymph nodes.

Our findings indicate that herd size has no influence on the incidence of *M. avium* in pig lymph nodes. This could be due to the fact that pig breeding in Poland is not consolidated, and the investigated animals mostly came from small farms of similar zoohygienic status. Nevertheless, it should be remembered that *M. avium* infections also occur at large farms with better zoohygienic conditions focused on intensive pig production [1, 7]. In addition, no correlation was observed between the incidence of *M. avium* and neither the nature of the lesions, nor the sex of the pigs. Similar findings have been noted for a population of wild boars from southern Spain, which also showed no links between the animal sex and infection rate [22].

The incidence of tuberculosis-like lesions in pig lymph nodes in the present study was found to be 0.81%. Similar results were achieved in a study conducted in two slaughterhouses

in the Netherlands, in which 0.75% of animals presented granulomatous lesions in the sub-mandibular nodes, and were located in mesenteric nodes in only one case. Our present findings indicate a much higher MAC isolation rate than in Dutch farms aimed at intensive production; this difference may be due to the susceptibility of the pig breed to infection, the nature of the animal husbandry system, the ecology of the bacteria [23], or the age of the pigs.

The prevalence of *M. avium* reached 56.98% in the lymph nodes with lesions, but only 19.47% in the unchanged ones. The relatively high incidence of *M. avium* in unchanged lymph nodes can be explained by the short life span of the pigs, and the fact that the infection occurred too late for the lesions to properly develop [24]. Similar *M. avium* isolation rates were published by Pate et al., who isolated the bacterium in 47.3% of pigs with changed lymph nodes and *R. equi* in 3.9%.

## Conclusions

The most effective screening method for the presence of *M. avium* in pig lymph nodes is an assessment of the number of tuberculosis-like lesions present during post-mortem inspection. In addition, evaluating the percentage of the node area covered with lesions also appears to be a useful indicator. In both cases, the recommended method of assessment is visual inspection. For material in the "single lesions" category, we recommend the simultaneous use of both methods. It should however, remembered that the latter method, i.e. estimating the area of the node covered with lesions, is difficult in slaughterhouse conditions, and that the nature of the lesion does not definitely confirm the presence of *M. avium*.

## Acknowledgments

The authors would like to thank the owners of slaughterhouses who agreed to be sampled at their facility.

## Author Contributions

**Conceptualization:** Aleksandra Kaczmarkowska.

**Data curation:** Aleksandra Kaczmarkowska, Anna Didkowska, Sylwia Brzezińska.

**Formal analysis:** Aleksandra Kaczmarkowska, Daniel Klich, Piotr Kociuba.

**Funding acquisition:** Krzysztof Anusz.

**Investigation:** Aleksandra Kaczmarkowska, Anna Didkowska.

**Methodology:** Aleksandra Kaczmarkowska, Daniel Klich.

**Resources:** Aleksandra Kaczmarkowska, Sylwia Brzezińska.

**Software:** Piotr Kociuba.

**Supervision:** Anna Didkowska, Izabella Dolka, Magdalena Rzewuska, Ewa Augustynowicz-Kopeć, Krzysztof Anusz.

**Validation:** Aleksandra Kaczmarkowska, Ewelina Kwiecień.

**Visualization:** Aleksandra Kaczmarkowska.

**Writing – original draft:** Aleksandra Kaczmarkowska.

**Writing – review & editing:** Aleksandra Kaczmarkowska, Krzysztof Anusz.

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
