## [Decision Letter · Decision Letter 0]

2 Feb 2022

PONE-D-21-34537Could the type and severity of gross lesions in pig lymph nodes play a role in the detection of *Mycobacterium avium?**PLOS ONE*

*Dear Dr. Kaczmarkowska*,

*Thank you for submitting your manuscript to PLOS ONE. After careful consideration, we feel that it has merit but does not fully meet PLOS ONE’s publication criteria as it currently stands. Therefore, we invite you to submit a revised version of the manuscript that addresses the points raised during the review process*.

*ACADEMIC EDITOR: As one of the reviewers have pointed out, this article needs robust and clear statistical section and inclusion of all p-values and their significance. This is particularly important since only a fraction of the total tested samples showed positivity for mycobacteria, and much fewer samples were confirmed as specific mycobacterial species. Secondly, the authors should clearly highlight and emphasize the strength/uniqueness as well as the limitations, such as the quality of the samples analyzed etc., on the manuscript. Finally, the article needs to be revised to remove redundant and irrelevant information in all sections, and  should focus  more on the main findings and their interpretations.*

*==============================*

*Please submit your revised manuscript by Mar 19 2022 11:59PM. If you will need more time than this to complete your revisions, please reply to this message or contact the journal office at plosone@plos.org. When you're ready to submit your revision, log on to https://www.editorialmanager.com/pone/ and select the 'Submissions Needing Revision' folder to locate your manuscript file*.

*Please include the following items when submitting your revised manuscript:*

*A rebuttal letter that responds to each point raised by the academic editor and reviewer(s). You should upload this letter as a separate file labeled 'Response to Reviewers'.*

*A marked-up copy of your manuscript that highlights changes made to the original version. You should upload this as a separate file labeled 'Revised Manuscript with Track Changes'.*

*An unmarked version of your revised paper without tracked changes. You should upload this as a separate file labeled 'Manuscript'.*

**

*If applicable, we recommend that you deposit your laboratory protocols in protocols.io to enhance the reproducibility of your results. Protocols.io assigns your protocol its own identifier (DOI) so that it can be cited independently in the future. For instructions see: https://journals.plos.org/plosone/s/submission-guidelines#loc-laboratory-protocols. Additionally, PLOS ONE offers an option for publishing peer-reviewed Lab Protocol articles, which describe protocols hosted on protocols.io. Read more information on sharing protocols at https://plos.org/protocols?utm_medium=editorial-email&utm_source=authorletters&utm_campaign=protocols.*

*We look forward to receiving your revised manuscript.*

*Kind regards,*

*Selvakumar Subbian, Ph.D.*

*Academic Editor*

*PLOS ONE*

*Journal Requirements:*

**

*Reviewers' comments:*

*Reviewer's Responses to Questions*

*

**Comments to the Author**
*

*1. Is the manuscript technically sound, and do the data support the conclusions?*

*The manuscript must describe a technically sound piece of scientific research with data that supports the conclusions. Experiments must have been conducted rigorously, with appropriate controls, replication, and sample sizes. The conclusions must be drawn appropriately based on the data presented. *

*Reviewer #1: Partly*

*Reviewer #2: Yes*

*2. Has the statistical analysis been performed appropriately and rigorously? *

*Reviewer #1: No*

*Reviewer #2: Yes*

*3. Have the authors made all data underlying the findings in their manuscript fully available?*

*The PLOS Data policy requires authors to make all data underlying the findings described in their manuscript fully available without restriction, with rare exception (please refer to the Data Availability Statement in the manuscript PDF file). The data should be provided as part of the manuscript or its supporting information, or deposited to a public repository. For example, in addition to summary statistics, the data points behind means, medians and variance measures should be available. If there are restrictions on publicly sharing data—e.g. participant privacy or use of data from a third party—those must be specified.*

*Reviewer #1: Yes*

*Reviewer #2: Yes*

*4. Is the manuscript presented in an intelligible fashion and written in standard English?*

*PLOS ONE does not copyedit accepted manuscripts, so the language in submitted articles must be clear, correct, and unambiguous. Any typographical or grammatical errors should be corrected at revision, so please note any specific errors here.*

*Reviewer #1: Yes*

*Reviewer #2: Yes*

*5. Review Comments to the Author*

*Please use the space provided to explain your answers to the questions above. You may also include additional comments for the author, including concerns about dual publication, research ethics, or publication ethics. (Please upload your review as an attachment if it exceeds 20,000 characters)*

*Reviewer #1: The paper entitled “Could the type and severity of gross lesions in pig….” by Kaczmarkowska et al. describes the development/validation of a simple tool for the rapid diagnosis of Mycobacterium avium complex members useful for the vets employed at slaughtering facility. I have appreciated the efforts put by the authors to validate such a kind of tool, but, despite the statistical approaches used in my opinion are in general quite correct (sometimes some values, e. g. Intervals of confidence and p-values are missing), an answer whit e certain degree of confidence about the presence or less of MAC members is far to be obtained, so it would be necessary in any case a more complex and expansive diagnostic assay to obtain a certain grade of confidence about the status of this specimens. Nevertheless I believe this kind of paper could be of some utility for those specifically employed in the slaughtering activity, so I encourage the authors to shorten the paper and to resubmit it in a new short version. Particularly the discussion section contains a lot of redundant or similar concepts that weigh down the text*

*Please, find here a few suggestions:*

*Row 59: I would say instead of “a collection” a “group”*

*Row 84: I would add a suggestion more appropriated , like the current European Union rules dedicated to this issue.*

*Row 149 and elsewhere: please report the centrifuge force as “g” and not as “rpm”.*

*Row 152: how many microliters of sediments have been inoculated?*

*Rows 185 187: suppose this model is based on the assumptions referred to the Ockham's razor? Could the author try to better explain this concept?*

*Row 251: there are any IC95% or similar? Maybe also correlated with a p-value?*

*Row 254-258: maybe here a further explanation about the model used could allow to the reader to better follow the explanation about low or high probability.*

*Row 268 Figure 6: It would be interesting if the authors report also the intervals of confidence for the model proposed.*

*Rows 300-301: Why PCR results here?*

*Rows 354-360: it sounds a little bit repetitive.*

*Reviewer #2: This study examined the size and type of tuberculous-like lymph node lesions found in 10,600 pigs at a slaughterhouse. The lymph nodes from 86 mandibular nodes with visible macroscopic changes and 113 without changes were further studied by examining a cross-section of the nodes and culturing M. avium on Lowenstein-Jensen and Stonebrink media and genotyping M. avium species using a commercial GenoType CM test. M. avium was found in 57% of lymph nodes with lesions but only 19% in unchanged lymph nodes suggesting a simple way to screen pigs to prevent entry of contaminated pork into the food chain. The following questions and comments should be addressed.*

*1. It was mentioned that visual inspection of the mandibular lymph nodes, as these appear to present with more frequent changes than mesenteric lymph nodes, by meat inspectors could serve to be a rapid and cost effective procedure for screening for macroscopic lesions (lines 287-298). How easy is it to access the mandibular lymph nodes to perform the inspection, i.e., how much time is involved to visually inspect these lymph nodes (minutes)?*

*2. Two subspecies of M. avium appear to be dominant in infecting pig lymph nodes, that being M. avium subspecies avium and M. avium subspecies hominissuis. Is there any difference in prevalence for these two types in infecting pigs and do both of these subspecies present a potential danger to humans? It was mentioned that humans may be infected with localized and pulmonary infections, particularly in immunocompromised individuals (lines 59-69). It was unclear from the GenoType CM test if it had the ability to distinguish between the subspecies avium and hominissuis as the Results only mention M. avium without differentiating the subspecies.*

*3. Staining included hematoxylin and eosin, presumably to assess lymphocytic infiltration common in granulomatous lesions. It was also mentioned that Kinyoun staining was used although it was not mentioned why (line 137). Kinyoun staining involving carbol fuchsin is generally used to stain mycobacteria red due to the presence of mycolic acid in order to detect acid fast bacilli. Was this the primary purpose for this staining and does M. avium stain positive with this technique?*

*Minor points*

*Line 216 “Necrosis separated by a thick one fibrous proliferation.” Unclear meaning*

*Line 392, “M. chelone” should be “M. chelonae”*

*6. PLOS authors have the option to publish the peer review history of their article (what does this mean?). If published, this will include your full peer review and any attached files.*

**

**

*Reviewer #1: No*

*Reviewer #2: **Yes: **John S. Spencer*

**

*While revising your submission, please upload your figure files to the Preflight Analysis and Conversion Engine (PACE) digital diagnostic tool, https://pacev2.apexcovantage.com/. PACE helps ensure that figures meet PLOS requirements. To use PACE, you must first register as a user. Registration is free. Then, login and navigate to the UPLOAD tab, where you will find detailed instructions on how to use the tool. If you encounter any issues or have any questions when using PACE, please email PLOS at figures@plos.org. Please note that Supporting Information files do not need this step.*

---

## [Author Response · Author response to Decision Letter 0]

24 Mar 2022

Dear Editor,

We would like to thank the Reviewers for providing such helpful comments. We have studied them carefully and all have been taken into consideration in our revision of the manuscript. All authors have approved of the revised manuscript and agree with the submission. 

Below we provide a detailed responses to the Reviewers’ comments.

We hope that the improved manuscript is acceptable for publication in PlosOne.

Yours faithfully,

Aleksandra Kaczmarkowska

Academic Editor: 

Comment: The article needs clear statistical section and inclusion of all p-values and their significance. This is particularly important since only a fraction of the total tested samples showed positivity for mycobacteria, and much fewer samples were confirmed as specific mycobacterial species. 

Answer: Thank you. We have added CIs and p-values and a new figure and provide a more extended explanation of the statistical differences. (Page 11, line 261-268, Figure 6)

Comment: The authors should clearly highlight and emphasize the strength/uniqueness as well as the limitations, such as the quality of the samples analyzed etc., on the manuscript. Answer: Thank you for your suggestion. We have added a brief description of the limitations of the method and have emphasized the uniqueness of this research in the Discussion. (Page 13, line 309-318)

Comment: The article needs to be revised to remove redundant and irrelevant information in all sections, and should focus more on the main findings and their interpretations.

Answer: Thanks for providing such helpful suggestions. We shortened the Discussion to remove irrelevant information and to highlight the main findings. We also extended the aims of the article to encompass the effect of gender and herd size on the incidence of tuberculosis-like lesions in pig lymph nodes.

Reviewer #1:

Comment: Sometimes some values, e. g. Intervals of confidence and p-values are missing

Answer: We agree: we have added CIs and p-values, as well as a new figure, and provide a more extended explanation of the statistical differences. Please see the answers to detailed comments below. (Page 11, line 261-268, Figure 6).

Comment: An answer whit e certain degree of confidence about the presence or less of MAC members is far to be obtained, so it would be necessary in any case a more complex and expansive diagnostic assay to obtain a certain grade of confidence about the status of this specimens.

Answer: Thank you for your valuable comment. We agree that a more complex study will provide a greater insight into the situation. However, we still believe that this short communication yields important conclusions for future research and has great scientific value in itself.

Comment: Shorten the paper and to resubmit it in a new short version. Particularly the discussion section contains a lot of redundant or similar concepts that weigh down the text

Answer: Thank you for your suggestion. We have revised the discussion and removed any irrelevant information. 

Comment: Row 59: I would say instead of “a collection” a “group”

Answer: Thank you for noticing. We have corrected it. (Page 3, line 59)

Comment: Row 84: I would add a suggestion more appropriated , like the current European Union rules dedicated to this issue.

Answer: Thank you for suggestion. We have corrected it. The relevant regulation of the European Commission was recalled. (Page 18, line 434-438)

Comment: Row 149 and elsewhere: please report the centrifuge force as “g” and not as “rpm”.

Answer: Thanks for noting this. We have changed the units from rpm to g. (Page 6, line 150; Page 6, line 152; Page 7, line 162)

 Comment: Row 152: how many microliters of sediments have been inoculated?

Answer: We have added this information in the Materials and Methods section. (Page 6, line 152)

Comment: Rows 185 187: suppose this model is based on the assumptions referred to the Ockham's razor? Could the author try to better explain this concept?

Answer: Thank you for your noticing. Yes, we followed the Ockham's razor for the generalized linear binary model selection, and this is a classic approach (according to Burnham and Anderson), where all model permutations are verified according the AIC values. We have provided a more extended explanation in the Methods. As the qualitative severity of the lesions in the lymph nodes, sex of the pigs and herd size could not explain the presence of M. avium in the samples, we analyzed only a percentage of the lesions in a logistic regression model. We hope that our explanations have clarified the description. (Page 8, line 190-193)

Comment: Row 251: there are any IC95% or similar? Maybe also correlated with a p-value?

Answer: Thank you for your suggestion. To clarify the results, we added Figure 6, presenting the frequency of M. avium against the severity of lesions, with CI and p values from the pairwise comparison. We have also added an explanation to the table about the reference category of no lesions. (Figure 6, Table 2)

Comment: Row 254-258: maybe here a further explanation about the model used could allow to the reader to better follow the explanation about low or high probability.

Answer: Following the comment, we have extended the appropriate description, including p-values of the pairwise comparison between the severity categories, and a figure showing the mentioned frequency of M. avium in lymph nodes. The pairwise comparison was performed based on the generalized binary model results (post hoc). (Page 11, line 261-268)

Comment: Row 268 Figure 6: It would be interesting if the authors report also the intervals of confidence for the model proposed.

Answer: We added 95% confidence intervals to the figure and information in the figure caption, as suggested. The figure number has been changed (current number is 7) as a result of adding a new figure.

Comment: Rows 300-301: Why PCR results here?

Answer: Thank you for your feedback. GenoType Mycobacterium CM should be used instead of RT-PCR. Of course, this procedure could be implemented if the costs were not important. (Page 14, line 322-323)

Comment: Rows 354-360: it sounds a little bit repetitive.

Answer: We agree – we have removed this part of the article. 

Reviewer #2:

Comment: It was mentioned that visual inspection of the mandibular lymph nodes, as these appear to present with more frequent changes than mesenteric lymph nodes, by meat inspectors could serve to be a rapid and cost effective procedure for screening for macroscopic lesions (lines 287-298). How easy is it to access the mandibular lymph nodes to perform the inspection, i.e., how much time is involved to visually inspect these lymph nodes (minutes)?

Answer: Thank you for your suggestions. We have added the estimated time taken to examine the lymph nodes in the Discussion, as well as the reason why we chose these lymph nodes for testing. (Page 13, line 302-305)

Comment: Two subspecies of M. avium appear to be dominant in infecting pig lymph nodes, that being M. avium subspecies avium and M. avium subspecies hominissuis. Is there any difference in prevalence for these two types in infecting pigs and do both of these subspecies present a potential danger to humans? It was mentioned that humans may be infected with localized and pulmonary infections, particularly in immunocompromised individuals (lines 59-69). It was unclear from the GenoType CM test if it had the ability to distinguish between the subspecies avium and hominissuis as the Results only mention M. avium without differentiating the subspecies.

Answer: Thank you for this comment. Unfortunately, our study is not accurate enough to speculate on the difference in the incidence of M. avium avium and M. avium hominisuis, and that was not the purpose of the study. The GenoType Mycobacterium CM test only allows for the differentiation by species. (Page 1, line 46)

Comment: Staining included hematoxylin and eosin, presumably to assess lymphocytic infiltration common in granulomatous lesions. It was also mentioned that Kinyoun staining was used although it was not mentioned why (line 137). Kinyoun staining involving carbol fuchsin is generally used to stain mycobacteria red due to the presence of mycolic acid in order to detect acid fast bacilli. Was this the primary purpose for this staining and does M. avium stain positive with this technique?

Answer: Thank you for your suggestion. We have added staining targets with the selected methods. Kinyoun staining is intended to visualize the mycobacteria. This is a positive staining method. (Page 6, line 135-137)

Comment: Necrosis separated by a thick one fibrous proliferation.” Unclear meaning

Answer: We have corrected this phrase. (Page 9, line 223-224)

Comment: “M. chelone” should be “M. chelonae”

Answer: We have removed this part of the article.

Richard Ente Claros

Comment: Please clarify if the data are ethically or legally restricted. If so, please state the reason(s) why the data are restricted.

Answer: Thank you for your noticing. The ethical approval of the National Ethical Committee for Animal Experiments was not required to conduct the tests because the lymph nodes were collected as a part of the post-mortem inspection of pigs in the slaughterhouse. No animal was intentionally sacrificed for this study. 

The authors confirm that the ethical policies of the journal, as noted on the journal’s author guidelines page, have been adhered with accordance to DIRECTIVE 2010/63/EU OF THE EUROPEAN PARLIAMENT AND OF THE COUNCIL of 22 September 2010 on the protection of animals used for scientific purposes. The data is not ethically or legally restricted.

Comment: If the data are restricted by an ethics committee or an Institutional Review Board (IRB), then please state the name of the IRB or ethics committee imposing these restrictions.

Answer: Not applicable. 

Comment: Please provide the contact information to which requests for data may be sent. Preferably, this would be an institutional email address to your IRB or ethics committee. If you are unable to provide a non-author contact point capable of handling data access requests, please explain why.

Answer: I give the address of this Ethics Committee.

E-mail: lke@sggw.edu.pl

---

## [Decision Letter · Decision Letter 1]

1 Jun 2022

Could the type and severity of gross lesions in pig lymph nodes play a role in the detection of *Mycobacterium avium?*

*PONE-D-21-34537R1*

*Dear Dr. Kaczmarkowska,*

*We’re pleased to inform you that your manuscript has been judged scientifically suitable for publication and will be formally accepted for publication once it meets all outstanding technical requirements.*

*Within one week, you’ll receive an e-mail detailing the required amendments. When these have been addressed, you’ll receive a formal acceptance letter and your manuscript will be scheduled for publication.*

*An invoice for payment will follow shortly after the formal acceptance. To ensure an efficient process, please log into Editorial Manager at http://www.editorialmanager.com/pone/, click the 'Update My Information' link at the top of the page, and double check that your user information is up-to-date. If you have any billing related questions, please contact our Author Billing department directly at authorbilling@plos.org.*

*If your institution or institutions have a press office, please notify them about your upcoming paper to help maximize its impact. If they’ll be preparing press materials, please inform our press team as soon as possible -- no later than 48 hours after receiving the formal acceptance. Your manuscript will remain under strict press embargo until 2 pm Eastern Time on the date of publication. For more information, please contact onepress@plos.org.*

*Kind regards,*

*Selvakumar Subbian, Ph.D.*

*Academic Editor*

*PLOS ONE*

* *

*Additional Editor Comments (optional):*

* *

*Reviewers' comments:*

*Reviewer's Responses to Questions*

*

**Comments to the Author**
*

*1. If the authors have adequately addressed your comments raised in a previous round of review and you feel that this manuscript is now acceptable for publication, you may indicate that here to bypass the “Comments to the Author” section, enter your conflict of interest statement in the “Confidential to Editor” section, and submit your "Accept" recommendation.*

*Reviewer #2: All comments have been addressed*

*2. Is the manuscript technically sound, and do the data support the conclusions?*

*The manuscript must describe a technically sound piece of scientific research with data that supports the conclusions. Experiments must have been conducted rigorously, with appropriate controls, replication, and sample sizes. The conclusions must be drawn appropriately based on the data presented. *

*Reviewer #2: Yes*

*3. Has the statistical analysis been performed appropriately and rigorously? *

*Reviewer #2: Yes*

*4. Have the authors made all data underlying the findings in their manuscript fully available?*

*The PLOS Data policy requires authors to make all data underlying the findings described in their manuscript fully available without restriction, with rare exception (please refer to the Data Availability Statement in the manuscript PDF file). The data should be provided as part of the manuscript or its supporting information, or deposited to a public repository. For example, in addition to summary statistics, the data points behind means, medians and variance measures should be available. If there are restrictions on publicly sharing data—e.g. participant privacy or use of data from a third party—those must be specified.*

*Reviewer #2: Yes*

*5. Is the manuscript presented in an intelligible fashion and written in standard English?*

*PLOS ONE does not copyedit accepted manuscripts, so the language in submitted articles must be clear, correct, and unambiguous. Any typographical or grammatical errors should be corrected at revision, so please note any specific errors here.*

*Reviewer #2: Yes*

*6. Review Comments to the Author*

*Please use the space provided to explain your answers to the questions above. You may also include additional comments for the author, including concerns about dual publication, research ethics, or publication ethics. (Please upload your review as an attachment if it exceeds 20,000 characters)*

*Reviewer #2: (No Response)*

*7. PLOS authors have the option to publish the peer review history of their article (what does this mean?). If published, this will include your full peer review and any attached files.*

**

**

*Reviewer #2: **Yes: **John S. Spencer*

---

## [Editor Report · Acceptance letter]

7 Jul 2022

PONE-D-21-34537R1 

Could the type and severity of gross lesions in pig lymph nodes play a role in the detection of *Mycobacterium avium*? 

Dear Dr. Kaczmarkowska:

I'm pleased to inform you that your manuscript has been deemed suitable for publication in PLOS ONE. Congratulations! Your manuscript is now with our production department. 

Kind regards, 

on behalf of

Dr. Selvakumar Subbian 

Academic Editor

PLOS ONE